# A fast and specific fluorescent probe for thioredoxin reductase that works via disulphide bond cleavage

Xinming Li[1], Baoxin Zhang[1], Chaoxian Yan[1], Jin Li[1], Song Wang[1], Xiangxu Wei[1], Xiaoyan Jiang[1], Panpan Zhou[1] & Jianguo Fang [1]

Small molecule probes are indispensable tools to explore diverse cellular events. However, finding a specific probe of a target remains a high challenge. Here we report the discovery of Fast-TRFS, a specific and superfast fluorogenic probe of mammalian thioredoxin reductase, a ubiquitous enzyme involved in regulation of diverse cellular redox signaling pathways. By systematically examining the processes of fluorophore release and reduction of cyclic disulfides/diselenides by the enzyme, structural factors that determine the response rate and specificity of the probe are disclosed. Mechanistic studies reveal that the fluorescence signal is switched on by a simple reduction of the disulfide bond within the probe, which is in stark contrast to the sensing mechanism of published probes. The favorable properties of Fast-TRFS enable development of a high-throughput screening assay to discover inhibitors of thioredoxin reductase by using crude tissue extracts as a source of the enzyme.

[1] State Key Laboratory of Applied Organic Chemistry & College of Chemistry and Chemical Engineering, Lanzhou University, Lanzhou 730000, China. Correspondence and requests for materials should be addressed to J.F. (email: fangjg@lzu.edu.cn)

Development of efficient and reliable tools is essential, yet a long-standing challenge, to explore the fundamental biological processes. A particular useful class of tools is the small molecule fluorescent probes, and this has driven an increasing interest to design and develop diverse chemical probes for biomacromolecules, inorganic ions, biological small molecules, and various reactive signaling species[1–6]. In general, the quality of a probe is critical for interrogating the complex biological systems and drawing convincing conclusions from experimental observations as probes with poor quality may generate ambiguous or even misleading results and conclusions[7]. Ideally, a desired probe should recognize a target of interest with an appropriate response rate and high specificity, which would guarantee the probe to be applied with confidence to give a precise and real-time dissection of a certain biological event.

Mammalian thioredoxin reductase (TrxR) enzymes are a family of selenoproteins that harbor a unique yet essential selenocysteine (Sec) residue at their C-terminal redox center[8]. TrxR, mainly through supplying electrons from NADPH to maintain the endogenous substrates thioredoxin (Trx) proteins in a reduced state, regulates a variety of redox-based signaling pathways that are involved in antioxidant defense, protein repair, and transcription regulation[9,10]. In our previous work, we reported a series fluorescent probes of TrxR, including TRFS-green, TRFS-red and Mito-TRFS (Fig. 1a)[11–13]. The practical applications of TRFS probes in multiple types of cells have facilitated the studies of TrxR, which have been demonstrated by not only our group but also researchers from other groups[14–21]. TRFS-green is the first fluorescent probe of TrxR with a green emission. However, the response rate of TRFS-green to the enzyme is slow, and it takes more than 2 h to reach the maximal fluorescence signal even using tris(2-carboxyethyl)phosphine (TCEP) as a reducing agent. In addition, the fold of fluorescence increase is moderate (~30-fold)[13]. Later, TRFS-red was reported with improved properties, such as a faster response rate (~1.5 h) and higher elevation of the fluorescence signal (~90-fold)[11]. Both probes showed good selectivity to TrxR over other related biological species, such as glutathione (GSH) and protein thiols.

In this work, we systematically investigate the structural factors that determine the response rate and selectivity of the probes via tuning their linker units and recognition parts (Fig. 1b). After optimizing the structure, a superfast probe, Fast-TRFS, is disclosed with a >150-fold increase of the emission intensity. Fast-TRFS reaches the maximal fluorescence signal within 1 min incubated with TCEP, and within 5 min incubated with the TrxR enzyme. More importantly, Fast-TRFS displays better selectivity to TrxR than do the TRFS-green and TRFS-red. Comparisons of TRFS-green, TRFS-red, and Fast-TRFS are summarized in Table 1. Further mechanistic studies reveal that switching on the fluorescence of Fast-TRFS is achieved by the reduction of the disulfide bond only (Fig. 1c), which is different

**Table 1 Comparisons of TRFS-green, TRFS-red and Fast-TRFS**

| Properties | Previous work | | This work |
|---|---|---|---|
| | TRFS-green | TRFS-red | Fast-TRFS |
| Response rate (TCEP, 1 mM) | +<br>(>2 h) | ++<br>(>1 h) | +++<br>(<1 min) |
| Response rate (TrxR, 50 nM) | +<br>(>3 h) | ++<br>(>1.5 h) | +++<br>(~5 min) |
| $F/F_0$ (TCEP, 1 mM) | +<br>(~30) | ++<br>(~90) | +++<br>(>150) |
| Selectivity (TrxR/GSH) | ++<br>(15.6) | +<br>(12.8) | +++<br>(55.7) |
| Ex & Em (nm) | ++<br>(438/538) | +++<br>(615/661) | +<br>(345/460) |
| Sensing mechanism | Cleavage-CDR | Cleavage-CDR | Cleavage only |

+: good; ++: better; +++: best

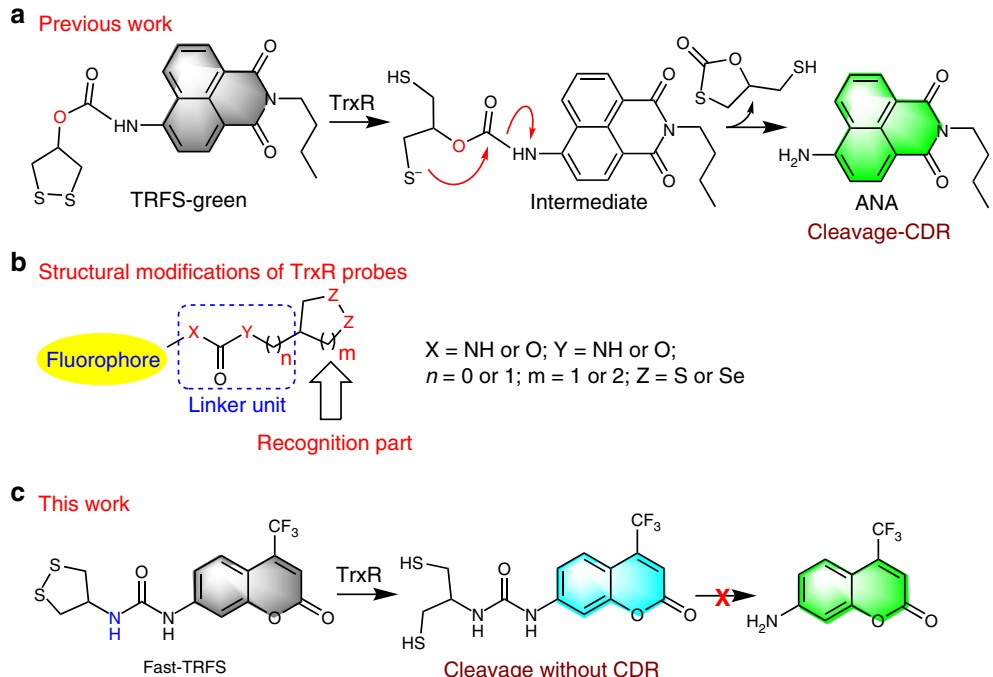

**Fig. 1** Reaction mechanism of TRFS-green and summary of current work. **a** Stepwise activation of TRFS-green. **b** Strategy for modification of TRFS-green. **c** One-step activation of Fast-TRFS

from the mechanism underlying the activation of TRFS-green or TRFS-red. The linker unit studied in this work is a well-established structure in constructing probes, prodrugs and theranostic agents via a process of the cyclization-driven release (CDR), and the disclosure of the structural determinants of such process would advance designing different molecules with improved properties. In addition, our clarification of the selective reduction of the 1, 2-dithiolanes by TrxR provides a general scaffold for constructing various chemical tools specifically targeting TrxR. Furthermore, the sensing mechanism of Fast-TRFS suggests that cleavage of the disulfide/diselenide bond may serve a direct trigger in design of fluorescent probes. Finally, with the aid of Fast-TRFS, a convenient assay is developed to screen TrxR inhibitors using crude cell extracts as a source of TrxR, and dozens of natural products have been identified as TrxR inhibitors.

## Results

**Detailed mechanisms of TRFS-green activation.** One major unfavorable property of TRFS-green is its slow response to TrxR[13]. As shown in Fig. 1a, there are two steps in the activation of TRFS-green. One is the generation of the intermediate via cleavage of the disulfide bond, and the other is a CDR process to liberate the aminonaphthalimide (ANA) fluorophore. The reaction process of TRFS-green and TCEP was monitored by high-performance liquid chromatography (HPLC) with a mass detector or photodiode array (PDA) detector (Fig. 2a, b), and our results showed clearly that the cyclization of the intermediate (the second step in Fig. 1a) contributes to the sluggish response of the probe. The detailed description of these results was given in the Supplementary Information. We also determined the time-dependent emission signal of the reaction mixture (Fig. 2c). Consistent with the steady increase of ANA production in the

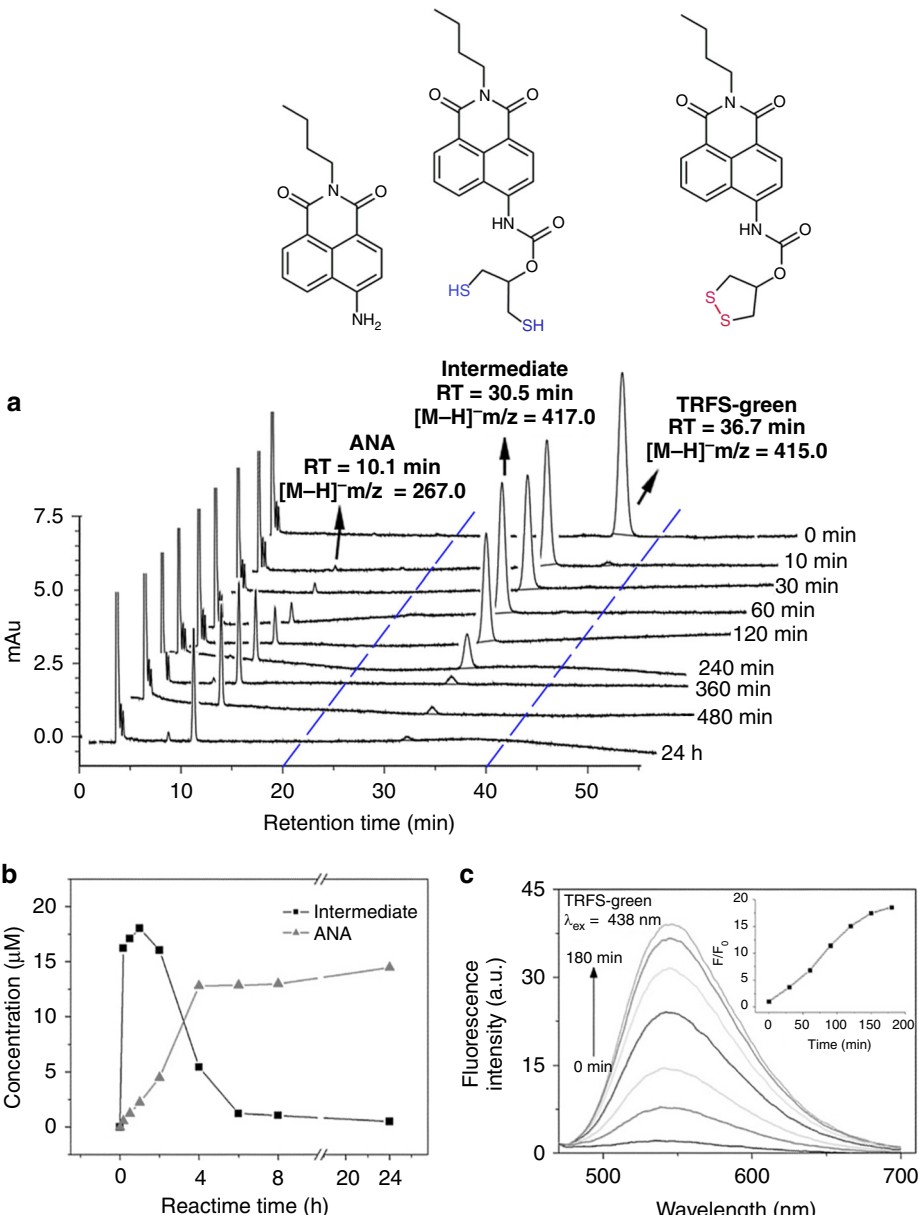

**Fig. 2** Stepwise activation of TRFS-green by TCEP. **a** TRFS-green (20 μM) was incubated with TCEP (1 mM) in TE buffer (50 mM Tris-HCl, 1mM EDTA, pH 7.4) at 37 °C for 24 h. The reaction mixture was analyzed by HPLC, and the quantification of intermediate and ANA was shown in (**b**). **c** Time-dependent fluorescence changes of TRFS-green (10 μM) in the presence of TCEP (1 mM) in TE buffer at 37 °C. The inset shows the time-dependent changes of emission at 538 nm ($\lambda_{ex}$ = 438 nm). Source data are provided as a Source Data file.Source Data file

**Fig. 3** Chemical Structures of TRFS series probes. The chemical structures of TRFS1-8 were presented

**Table 2 Summary of optical properties of TRFS1-8[a]**

| Probes | TCEP (F/F$_0$) | | GSH | TrxR |
|---|---|---|---|---|
| | $\lambda_{ex} = 438$ nm $\lambda_{em} = 538$ nm | $\lambda_{ex} = 365$ nm $\lambda_{em} = 495$ nm | | |
| TRFS-green | >18 (3 h) | ND | − | + |
| TRFS1 | Unstable | Unstable | ND | ND |
| TRFS2 | Unstable | Unstable | ND | ND |
| TRFS3 | − | >10 (2 min) | − | + |
| TRFS4 | ~4 (4 h) | ND | − | + |
| TRFS5 | >80 (10 min) | ND | + | + |
| TRFS6 | − | >100 (5 min) | − | − |
| TRFS7 | ~17 (3 h) | ND | − | − |
| TRFS8 | − | ~3 (15 min) | − | − |

ND not determined, +: having fluorescence signal, −: no significant fluorescence signal
[a]The assays were performed by incubating the probes (10 μM) with TCEP (1 mM), GSH (1 mM) or TrxR/NADPH (50 nM and 200 μM), and fluorescence spectra were recorded

first 4 h, the fluorescence signal kept increasing during the period of detection.

**Strategies to modify TRFS-green.** Since we have demonstrated that the cyclization step is responsible for the slow response of TRFS-green, we then hypothesized that modifying the structure of TRFS-green to improve the rate of cyclization might create different probes with improved response rate. We chose the naphthalimide scaffold as a fluorophore, and prepared a series of potential probes with varying linker units and recognition parts (TRFS1-8, Fig. 3). The detailed synthetic procedures and characterization of these probes were described in the Supplementary Information. TRFS1-3 were designed to examine the effect of the leaving group (O vs N) on the release of the fluorophore. TRFS4 was designed to examine whether forming a six-membered thiocarbonate favors the release of the fluorophore. TRFS5 and TRFS6 were prepared to study the nucleophilic attack of the carbonyl group by selenolate. TRFS7 and TRFS8 (and also TRFS5 and TRFS6) were designed to examine whether the recognition parts other than the 1,2-dithiolane unit could be used in constructing TrxR probes. In our attempt to synthesize TRFS1 and TRFS2, we noticed that both compounds were not stable. TRFS1 decomposed during work-up of the reaction mixture, and TRFS2 hydrolyzed spontaneously in aqueous buffer giving the corresponding fluorophore (hydroxynaphthalimide). Thus both TRFS1 and TRFS2 were excluded in the following experiments.

**Screening of TRFS probes.** TCEP is a strong reducing agent, and may readily reduce disulfide and diselenide bonds. To simplify the experimental procedure, we employed TCEP in place of TrxR/NADPH for the initial screening of the probes (TRFS3-8). The results were summarized in Table 2 (the second and third columns), and the response of probes to GSH and TrxR was summarized in the last two columns. The detailed description of these results was given in the Supplementary Information. Based on these results, a preliminary structure-activity relationship (SAR) of these probes in responding to TCEP was drawn: 1) The

linker atom directly connecting to the fluorophore (X in Fig. 1b) determines the stability of the probes, and the nitrogen atom is expected to improve the stability while probes contain the oxygen atom are not stable (TRFS1 and TRFS2); 2) Both the Y atom and the number of carbons between the recognition part and the Y atom (Y and n in Fig. 1b) determine the fluorophore release. To efficiently liberate a fluorophore, Y = O and $n = 0$ are preferred (TRFS-green, TRFS5, and TRFS7); 3) Replacement of the disulfide in the recognition part with diselenide promotes fluorophore release, but the selectivity was compromised (TRFS5 & TRFS-green); 4) The recognition part containing five-membered cyclic disulfides showed selectivity for TrxR over GSH. As the CDR strategy (the second step of TRFS-green activation shown in Fig. 1a) is widely applied in designing probes, prodrugs and theranostic agents[22–32], our clarification of the SAR of such type molecules would have a general interest and could advance the development of different controlled release systems.

**Sensing mechanism of TRFS3.** As TRFS3 showed fast response to TCEP, and displayed selectivity for TrxR over GSH (Table 2), the detailed reaction process of TRFS3 with TCEP was monitored by HPLC coupled with a mass or PDA detector (Fig. 4a). Our results demonstrated that the disulfide bond in TRFS3 was cleaved quantitatively within 1 min, but no ANA was detected even extending the reaction to 4 h, indicating the following CDR process did not take place (Fig. 4b). This is likely due to the stability of the urea linker unit (-NH-C(O)-NH-), and the cyclization by the nascent thiolate attack was not favorable, and thus no ANA was released. The detailed description and interpretation of these results were given in the Supplementary Notes. Taken together, these results demonstrated that a direct reduction of TRFS3 without the following cyclization (Fig. 4b) occurred in the response of the probe to TCEP. Furthermore, the off-on fluorescence signal of TRFS3 (and other probes, such as TRFS6 and TRFS8, Table 2) in response to TCEP also suggested that the disulfide/diselenide bond could quench the emission of certain fluorophores, and may serve as a trigger in designing fluorescent probes.

**Reduction of cyclic disulfides and diselenides.** Discovery of small molecule ligands of a protein of interest is critical for chemical manipulation of the protein. Disulfides and diselenides are a class of redox-active compounds with multiple biological functions. It has been well documented that many linear disulfides/diselenides are good substrates of TrxR[33–37]. However,

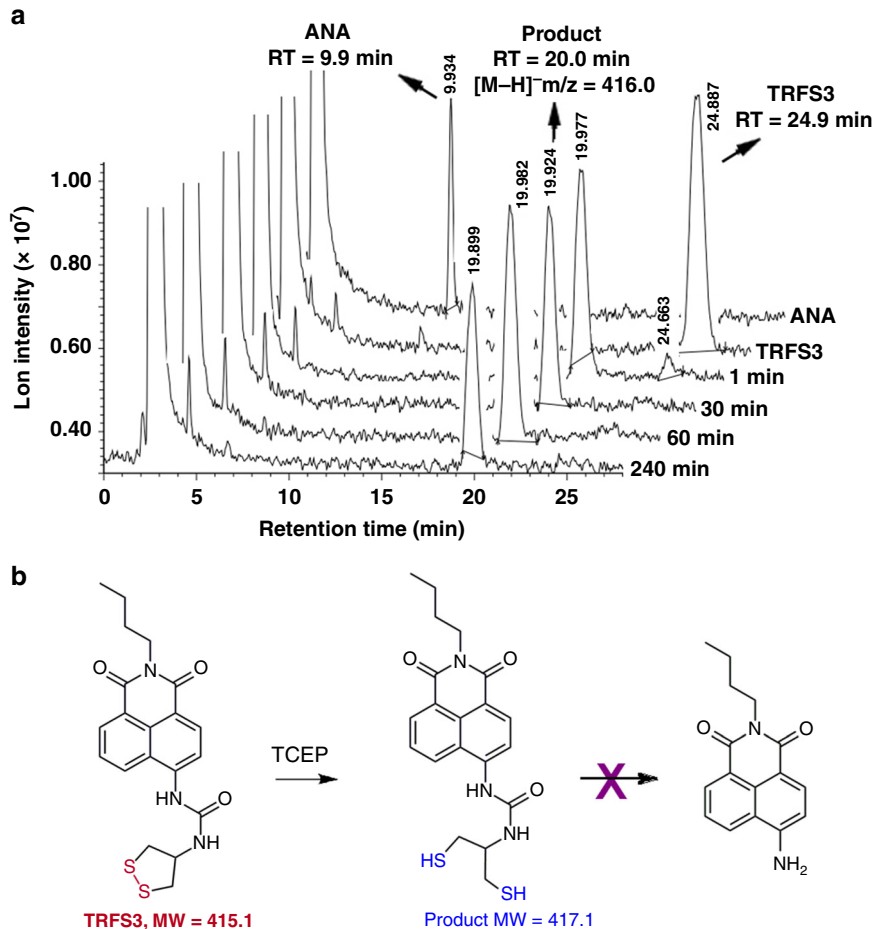

**Fig. 4** Reaction details of TRFS3 and TCEP. **a** TRFS3 (20 μM) was incubated with TCEP (1 mM) in TE buffer at 37 °C for 4 h. The reaction mixture was analyzed by HPLC-MS. **b** Proposed mechanism for the reduction of TRFS3 by TCEP. Source data are provided as a Source Data file.

**Table 3 Reduction of cyclic disulfides/diselenides by TrxR and GSH[a]**

| Compounds | Blank | 1 | 2 | 3 | 4 | 5 | 6 | 7 | 8 | 9 |
|---|---|---|---|---|---|---|---|---|---|---|
| TrxR/ NADPH | 0.54 ±0.04 | 1.15 ±0.16 | 2.58 ±0.32 | 21.59 ±1.20 | 0.52 ±0.08 | 2.70 ±0.10 | 0.24 ±0.13 | 0.42 ±0.10 | 0.40 ±0.07 | 2.74 ±0.55 |
| GR/GSH/ NADPH | 0.70 ±0.05 | 0.80 ±0.08 | 0.95 ±0.10 | 1.03 ±0.21 | 1.74 ±0.45 | 3.42 ±0.24 | 0.88 ±0.02 | 0.78 ±0.04 | 1.12 ±0.05 | 2.07 ±0.40 |

[a]The assays were performed by incubating the compounds (100 μM) with the recombinant rat TrxR/NADPH (50 nM and 200 μM) or GSH (1 mM), GR/NADPH (0.5 U mL$^{-1}$ and 200 μM) in TE buffer for 10 min at 37 °C. The rates of NADPH decay were calculated based on the change of $A_{340}$ within the initial 3 min. The data were expressed as mean ± standard deviation (SD, $n = 3$). Source data are provided as a Source Data file

studies on the interaction of cyclic disulfides with TrxR are limited[38–41], and there is no study on the interaction of cyclic diselenides with TrxR. To extend this preliminary result, i.e., the selective reduction of 5-membered cyclic disulfides by TrxR, we further prepared a series of cyclic disulfides/diselenides (1–9, Table 3), and studied their interactions with TrxR and GSH. The detailed description and interpretation of these results were given in the Supplementary Notes. Based on the data in Table 3, the SAR of reduction of these molecules could be drawn. First, the 1, 2-dithianes (6-membered cyclic disulfides, compounds 6 and 7)

cannot be reduced by either TrxR or GSH; Second, the 1, 2-dithiolanes (5-membered cyclic disulfides, compounds 1, 2, and 3) are substrates of TrxR but cannot be reduced by GSH; Third, the reduction of the cyclic diselenides is a little bit complicated: Compounds 5 and 9 are substrates of both TrxR and GSH, while compound 8 is resistant to TrxR but appears a weak substrate of GSH. Interestingly, compound 4 seems to be selectively reduced by GSH but not by TrxR. Taken together, although more data are needed to obtain a clear picture of reduction of cyclic diselenides, it is evident that 1, 2-dithiolanes display promising selectivity to

TrxR over GSH, which strongly supports the selective activation of TRFS3 and TRFS4 by TrxR. This discovery demonstrated that the 1, 2-dithiolane moiety may serve a general ligand in designing various chemical tools to target TrxR selectively.

Our discovery of selective reduction of 1,2-dithiolanes by TrxR is also supported by the preceding studies. First, the reduction of 1, 2-dithiolane moiety in lipoic acid by TrxR was reported by Arner and coworkers[40], while chaetocin and gliotoxin, two fungal metabolites containing the 1, 2-dithiane moiety, were poor substrates of TrxR demonstrated by researchers from the Bible group and Kwon group, respectively[38,39]. Hondal et al. also confirmed that TrxR efficiently reduced lipoic acid, but had little effect on the oxidized dithiothreitol[41]. Second, the interactions of GSH/TrxR with disulfides/diselenides involve a general thiol/ selenol and disulfides/diselenides exchange reaction, which is ubiquitous in biological systems and forms a fundamental of the biological redox regulation[42,43]. There are two major factors, i.e., the nucleophilicity of the attacking group and the electrophilicity of the atom accepting the electrons from the nucleophile, to determine such exchange reactions[44,45]. Under neutral conditions, selenols are ~390-fold and ~130-fold more reactive than thiols in the exchange reactions with disulfides and diselenides, respectively[45]. Many natural selenoenzymes also employ the Sec residue to accelerate the thiol/disulfide-like exchange reactions[42]. Third, Singh and Whitesides reported that 1, 2-dithiolanes are approximate three orders of magnitude more reactive than 1,2-dithianes in the thiol/disulfide exchange reactions[46], which is due to the large ring strain in the 1, 2-dithiolane ring system. Intriguingly, Butora et al. reported 1, 2-dithiane-based prodrugs that can be activated by GSH[47]. They proposed that the first step of the prodrug activation involves the cleavage of the disulfide bond in 1,2-dithiane moiety by GSH. However, under our experimental conditions, we only observed a marginal reduction of 1,2-dithianes (compounds 6 and 7) by GSH. The previous work studying the direct reaction between GSH and oxidized dithiothreitol[48] and the recent work from the Tang group[25] also supported that GSH has little effect on the reduction of the 1, 2-dithiane moiety.

**Further design of TrxR probes**. When analyzing the fluorescence response of the TRFS1-8 to TCEP (Fig. 3 and Supplementary Fig. 1), we noticed that TRFS3, TRFS6, and TRFS8 displayed a fast response to TCEP with emission spectra ($\lambda_{ex}$ = 365 nm, $\lambda_{em}$ = 495 nm) distinct from that of the expected ANA fluorophore ($\lambda_{ex}$ = 438 nm, $\lambda_{em}$ = 538 nm). This indicated that no ANA was released after the cleavage of the disulfide or diselenide bonds in these molecules, which was further supported by the study of the reaction between TRFS3 and TCEP (Fig. 4). We were particularly interested in the remarkable elevation of the fluorescence signal within just a few minutes by simple cleavage of the disulfide/ diselenide bonds, as this might lead to a different mechanism in sensing TrxR. The three probes share the same linker unit (N & N), which is too stable to allow the following CDR process. Taking into consideration of the selective reduction of the 1, 2-dithiolane scaffold by TrxR, we next prepared a series of potential probes (TRFS9-13, Fig. 5a) with varying fluorophores. These probes keep the 1, 2-dithiolane scaffold as a recognition part. The N & N linker unit was used in TRFS9, TRFS12, and TRFS13. The stable amide and sulfonamide linkers were used in TRFS10 and TRFS11, respectively. The detailed synthetic procedures and characterization of TRFS9-13 were described in the Supplementary Information. With these probes in hands, we then screened their fluorescence signal upon addition of TCEP.

As shown in Fig. 5b-d, the fluorescence signals of TRFS9 ($\lambda_{ex}$ = 345 nm, $\lambda_{em}$ = 460 nm), TRFS10 ($\lambda_{ex}$ = 490 nm, $\lambda_{em}$ = 510 nm) and TRFS12 ($\lambda_{ex}$ = 324 nm, $\lambda_{em}$ = 460 nm) were readily triggered on by TCEP with a ~160-, ~4- and ~30-fold increase of the emission intensity, respectively. However, there was no significant elevation of the fluorescence signal for TRFS11 and TRFS13 upon addition of TCEP (Supplementary Fig. 11). It is worth noting that the response of TRFS9 to TCEP was superfast. The fluorescence signal reached the maximal intensity within 1 min, and was stable after incubating the probe with TCEP for 30 min (inset in Fig. 5b). The absolute quantum yield of TRFS9 was determined to be 3.3%, and it increased to 40.6% after reacting with TCEP. There is no obvious change of the absorbance spectra upon reduction of TRFS9 by TCEP (Supplementary Figure 12). Based

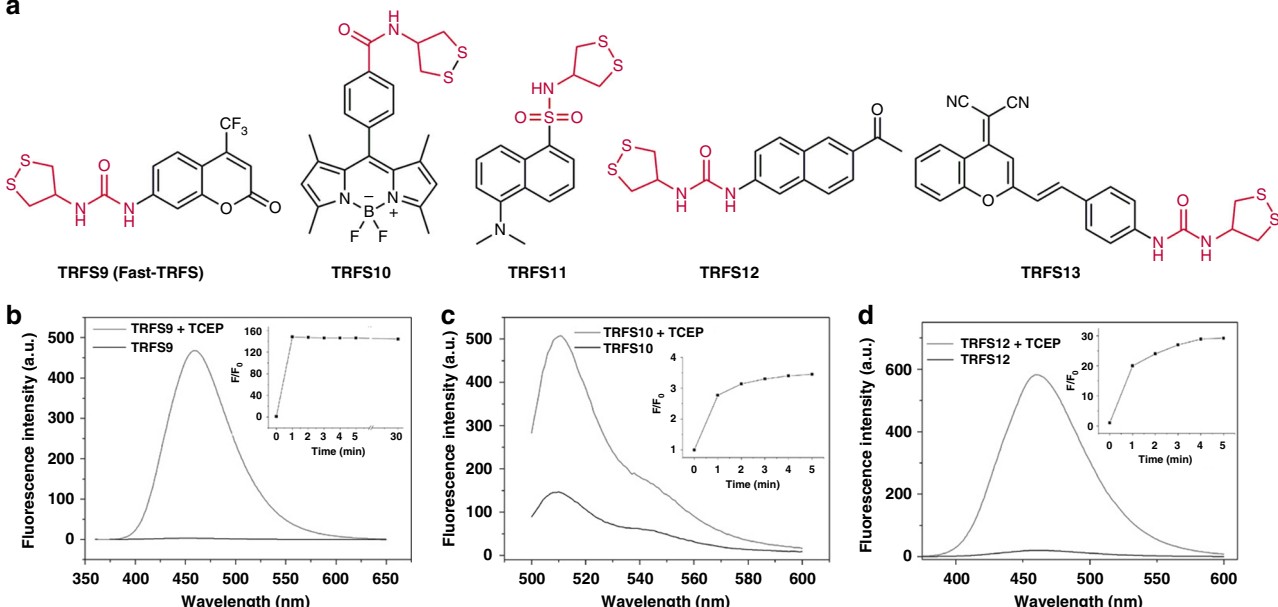

**Fig. 5** Structures of TRFS9-13 and their response to TCEP. **a** Structures of TRFS9-13. The probes (10 μM) were incubated with TCEP (1 mM) at 37 °C in TE buffer. The time-dependent emission spectra were recorded. **b** $\lambda_{ex}$ = 345 nm; **c** $\lambda_{ex}$ = 490 nm; **d** $\lambda_{ex}$ = 324 nm. The insets in **b–d** showed the time-dependent changes of emission at 460 nm, 510 nm, and 460 nm, respectively. Source data are provided as a Source Data file

on this initial observation and the favorable properties of TRFS9, we termed it as Fast-TRFS and decided to choose it for further evaluation.

**Sensing mechanism of Fast-TRFS (TRFS9).** First, we asked the reaction mechanism underlying the off-on fluorescence signal of Fast-TRFS elicited by TCEP. We employed the HPLC coupled with a mass detector to monitor the reaction of the probe with TCEP. Fast-TRFS was eluted with a RT of 12.21 min (Fig. 6a0). After addition of TCEP and incubation for 1 min, the probe peak almost disappeared completely and another peak with a RT of 10.27 min was observed (Fig. 6a1). The mass detector showed that the MW of this peak is 377.0 ($[M-H]^-$). Compared to the probe Fast-TRFS which has a MW of 375.0 ($[M-H]^-$), this peak indicated the generation of a reduced product of Fast-TRFS (R-Fast-TRFS, Fig. 6a). Further extending the reaction time to 30 min did not change the yield of the product (Fig. 6a30), which supported that the reduction of Fast-TRFS by TCEP is a fast process and was consistent with the changes of fluorescence spectra shown in Fig. 5b. A slight decrease of the R-Fast-TRFS and increase of Fast-TRFS were observed for longer incubation time (240 min, Fig. 6a240). This is likely due to the partial oxidation of the R-Fast-TRFS. These results further supported the aforementioned observations, i.e., the urea linker -NH-C(O)-NH- is stable to forbid the CDR process and cleavage of disulfide/diselenide bonds could light up fluorescence directly. More importantly, the hitherto unknown sensing mechanism via simple cleavage of disulfide/diselenide bonds would have a broad interest, and advance the creation of different sensing probes. The $^1$HNMR spectra of Fast-TRFS and R-Fast-TRFS were also determined (Supplementary Fig. 13). There are no apparent changes of

protons' signal from the coumarin skeleton and the –NH-C(O)-NH- linker unit. However, cleavage of the disulfide bond of Fast-TRFS affected the proton's signal from the five-membered ring moiety, and the detailed assignment of the proton's signal was shown in Supplementary Fig. 13.

To further understand the different emission properties of Fast-TRFS and R-Fast-TRFS, we performed theoretic calculations. For Fast-TRFS, it was calculated that the absorption spectrum of the molecule is principal from the excitation of its ground state (GS) to the singlet excited state S2 (HOMO-1 → LUMO transition, 89.4% contribution) (Supplementary Table 1). However, the S2→GS fluorescence emission (LUMO → HOMO-1 transition, 88.5% contribution) was predicted by calculation but not observed in experiments (Supplementary Table 3). The calculated S1→GS fluorescence emission (LUMO + 1 → HOMO transition) with extremely low oscillator strength (0.001 oscillator strength) should be responsible for the weak fluorescence observed in experiments. When Fast-TRFS is in the optimized S2 excited structure, the energy difference between S2 state and S1 state were only 0.22 eV. Thus, we reasoned that S2→S1 internal conversion occurred to overwhelm the direct S2→GS transition, which might account for the low emission from Fats-TRFS. In the case of R-Fast-TRFS, the calculation results indicated that both the absorption spectrum and fluorescence emission spectrum were from the HOMO-LUMO transition (GS←→S1, Supplementary Tables 2 and 4). The detailed description and interpretation of calculation results were given in the Supplementary Notes.

**Selective response of Fast-TRFS to TrxR.** We next determined the response of Fast-TRFS to TrxR in detail. When the probe was

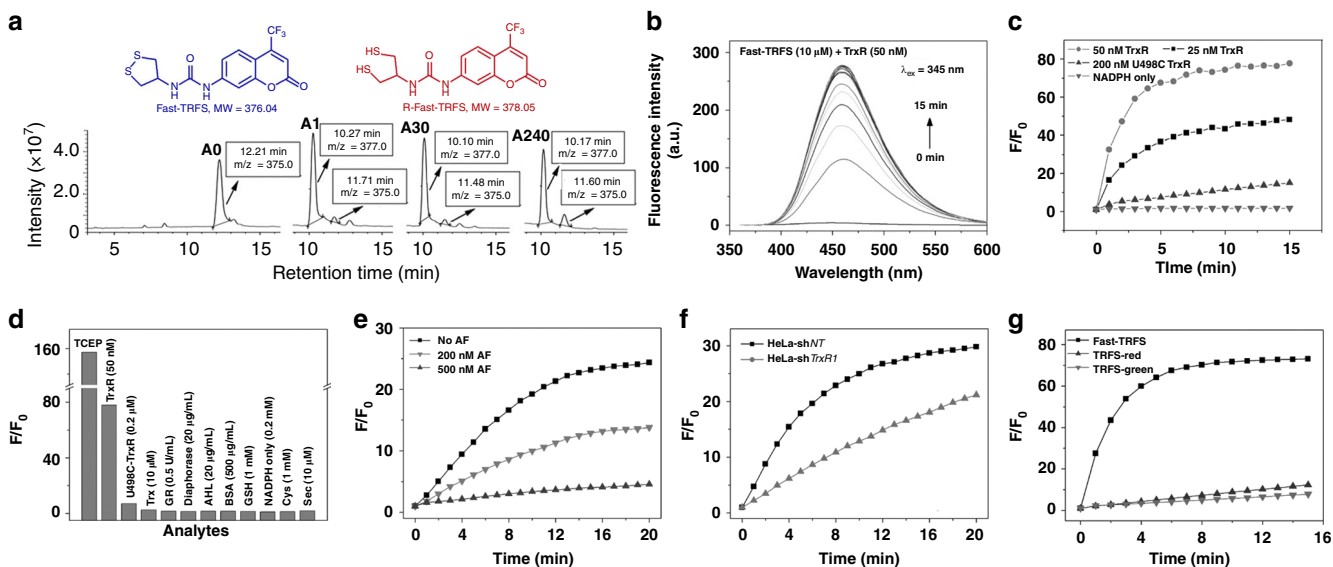

**Fig. 6** Reduction of Fast-TRFS (TRFS9) by TCEP and Selective activation of Fast-TRFS by TrxR. **a** Reduction of Fast-TRFS by TCEP. Fast-TRFS (100 μM) was incubated with TCEP (1 mM) for 0 min (**a**0), 1 min (**a**1), 30 min (**a**30) and 240 min (**a**240), and the mixture was analyzed by HPLC-MS. **b** Time-dependent emission spectra of Fast-TRFS toward TrxR. Fast-TRFS was incubated with TrxR/NADPH (50 nM/200 μM), and the emission spectra were recorded every 1 min for 15 min. **c** Time course of the fluorescence increase of Fast-TRFS with TrxR/NADPH and U498C TrxR/NADPH. **d** Response of Fast-TRFS to various relevant biological species. The fluorescence increase at 460 nm was determined after they were incubated with Fast-TRFS for 15 min. The Sec (10 μM) was generated in situ by mixing Cys (1 mM) and selenocystine (5 μM). **e** Inhibition of the cell lysate-mediated Fast-TRFS reduction by TrxR inhibitor AF. The NADPH-pretreated HeLa cell lysate (0.5 mg mL⁻¹) was treated with AF for 30 min, and further incubated with Fast-TRFS and NADPH (200 μM) for additional 20 min. The fluorescence increase was determined. **f** Reduction of Fast-TRFS by lysates (0.5 mg mL⁻¹) from the genetically manipulated HeLa cells in the presence of NADPH (200 μM). **g** Time course of the fluorescence increase of Fast-TRFS, TRFS-red and TRFS-green with TrxR/NADPH (50 nM/200 μM). All reactions were performed in TE buffer at 37 °C. The excitation/emission wavelengths for Fast-TRFS, TRFS-green and TRFS-red are 345/460 nm, 438/538 nm and 615/661 nm, respectively. The concentration of the probes in (**b**–**g**) was 10 μM. Source data are provided as a Source Data file

incubated with the recombinant TrxR (50 nM), the fluorescence spectra were recorded every minute for 15 min ($\lambda_{ex}$ = 345 nm, Fig. 6b). The maximal emission could be reached at around 5 min with a near 80-fold increase of the fluorescence intensity. The response of Fast-TRFS to varying concentrations of TrxR was shown in Fig. 6c. Interestingly, the Sec-deficient U498C TrxR, a mutant TrxR by replacing the 498th Sec to cysteine (Cys), gave a very weak fluorescence signal even at a high concentration (200 nM), indicating that the Sec residue in TrxR is essential in triggering on the fluorescence of Fast-TRFS. Other relevant species, such as Trx, glutathione reductase (GR), diaphorase, amidehydrolase (AHL), bovine serum albumin (BSA), GSH, NADPH, Cys and Sec, displayed little interference on the fluorescence signal (Fig. 6d). Increasing the GSH concentration to 10 mM also gave little fluorescence signal (Supplementary Figure 14). The probe showed a 56-fold selectivity for TrxR (50 nM) over the GSH (1 mM). Based on the previous published data[11,13], the selectivity of TRFS-green and TRFS-red for TrxR over GSH were calculated to be 15.6 and 12.8, respectively. The crude HeLa cell extracts also showed robust activity in eliciting the fluorescence from Fast-TRFS, and this activity was dose-dependently inhibited by a selective TrxR inhibitor auranofin (AF) (Fig. 6e), suggesting that the response of the probe to TrxR is highly selective. To further evaluate the specific response of Fast-TRFS to TrxR, we compared the reduction of the probe by lysates from two genetically manipulated HeLa cells, i.e., HeLa-sh*NT* cells and HeLa-sh*TrxR1* cells, which were prepared by transfecting HeLa cells with a non-targeting shRNA plasmid and a shRNA plasmid specifically targeting the *TrxR1*, respectively[16,18], The silence efficiency of the RNA interference was previously validated to be more than 70%. Total TrxR activity in the HeLa-sh*TrxR1* cell lysates was determined to be around 40% of that in HeLa-sh*NT* cell lysate by the

classic Trx-mediated endpoint insulin reduction assay. The HeLa-sh*NT* lysate showed similar ability as the normal HeLa cell lysate to activate the probe (Fig. 6e, f), but was much better than the HeLa-sh*TrxR1* lysate to reduce the probe (Fig. 6f), which strongly supported a specific activation of Fast-TRFS by TrxR. Comparisons of Fast-TRFS with the two previous reported TrxR probes, i.e., TRFS-green and TRFS-red, under the same conditions were shown in Fig. 6g. It is clear that Fast-TRFS had a much faster response to TrxR than the TRFS-green and TRFS-red. A more than 70-fold increase of the fluorescence intensity was observed for Fast-TRFS, while there was only less than 10-fold increase for either TRFS-green or TRFS-red within the first 10 min. Taken together, Fast-TRFS displayed multiple favorable properties, such as improved specificity, fast response and high amplitude of fluorescence increase, to respond to TrxR.

We then demonstrated that Fast-TRFS could image TrxR activity in live cells with high specificity. As shown in Fig. 7a, the blue fluorescence signal appeared within 2 min after addition of the probe into the cultured HeLa cells. This fluorescence increased as extending the incubation times, and could reach a saturated signal within 30 min. The intracellular fluorescence could be dose-dependently inhibited by the TrxR inhibitor AF (Fig. 7b). Quantification of the fluorescence intensity was performed on a fluorescence microplate reader, and the results were shown in Fig. 7d. More convincing evidence was from the comparison of the time-dependent changes of the fluorescence signal in HeLa-sh*NT* cells and HeLa-sh*TrxR1* cells (Fig. 7c). Quantification of the fluorescence intensity by a fluorescence microplate reader was shown in Fig. 7e. The fluorescence in the HeLa-sh*NT* cells was stronger than that in the HeLa-sh*TrxR1* cells, indicating that the HeLa-sh*NT* cells were more efficient to activate the probe than the HeLa-sh*TrxR1* cells. Taken together,

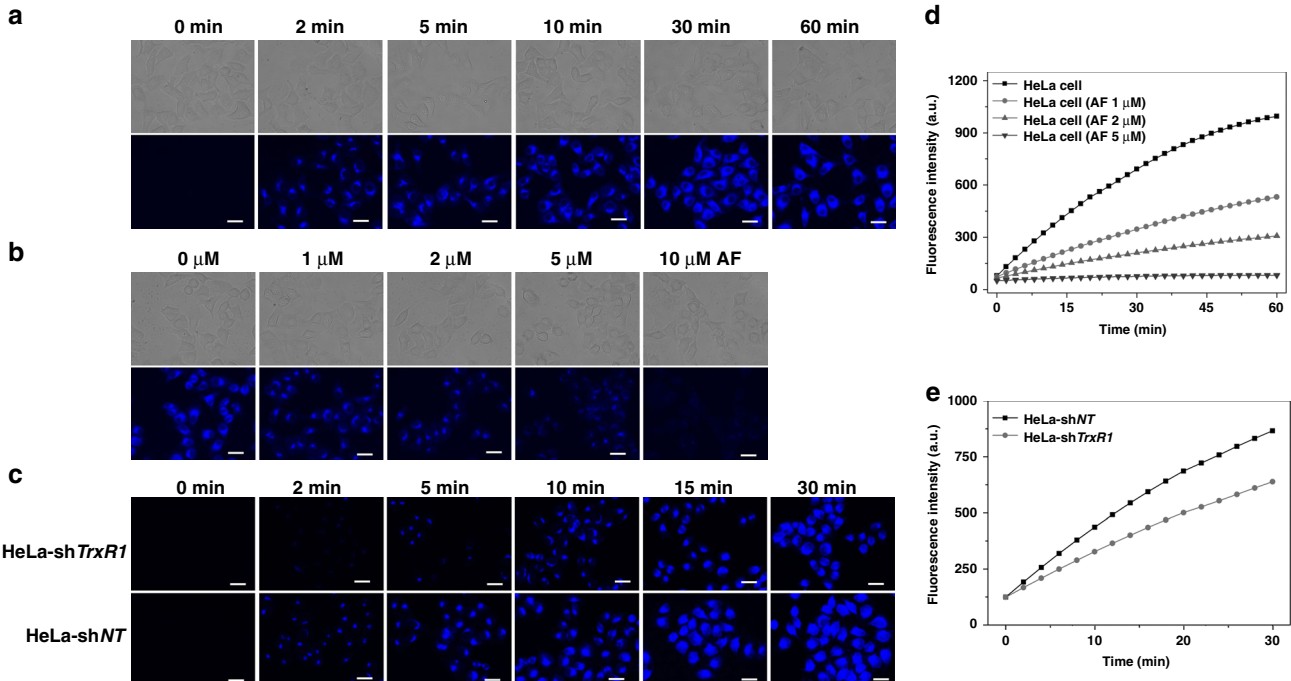

**Fig. 7** Activation of Fast-TRFS in live cells. **a** HeLa cells were treated with Fast-TRFS (10 μM) for the indicated times, and bright field and fluorescence images were acquired. **b** HeLa cells were pretreated with different concentrations of TrxR inhibitor AF for 2 h, and then were further stained with Fast-TRFS (10 μM) for 15 min. Bright field and fluorescence images were shown. **c** HeLa-sh*TrxR1* cells and HeLa-sh*NT* cells were treated with Fast-TRFS (10 μM) for the indicated times, and fluorescence images were shown. **d** Quantification of the fluorescence intensity from (**a**) and (**b**) by a fluorescence plate reader (Tecan Infinite M200). **e** Quantification of the fluorescence intensity from (**c**) by a fluorescence plate reader. Scale bars: 25 μm. Source data are provided as a Source Data file

pharmacological inhibition and genetic knockdown of the enzyme suppressed the activation of Fast-TRFS in live cells, demonstrating a specific activation of the probe by TrxR.

**TrxR assay development and TrxR inhibitors identification**. TrxR is a selenoenzyme. It is still a challenge to prepare recombinant selenoproteins from the prokaryotic expression system as the Sec is encoded in a special way by a UGA codon, which is normally a stop codon in protein translation. Since we have demonstrated that Fast-TRFS is a specific probe of TrxR, we hypothesized that the probe may be useful in screening TrxR inhibitors using the crude tissue protein extract as a source of TrxR. As shown in Fig. 6e, the HeLa cell lysate efficiently reduced the probe giving a time-dependent increase of the fluorescence signal, and this fluorescence signal was dose-dependently inhibited by the known TrxR inhibitor AF. With these results in hands, we adapted this cuvette-based method to a microplate assay to screen a self-built small library containing 72 natural compounds. The detailed structural information of all compounds was shown in Supplementary Figure 15. All the known TrxR inhibitors in the library, such as celastrol[49], securinine[50], plumbagin[21], alantolactone[16], parthenolide[18], xanthatin[15], gambogic acid[51], curcumin[52], piperlongumine[53], shikonin[54], cynaropicrin[55], baicalein[56], and myricetin[57] were identified, indicating the reliability of the Fast-TRFS-based screening assay. In addition, twenty inhibitors with > 20% inhibition of the enzyme activity under the screening conditions were discovered: embelin, tanshinone IIA, eupalinolide A, dehydroleucodine, eupalinilide B, gossypol, cynarin, isoliquiritigenin, fisetin, disulfiram, exemestane, cichoric acid, piceatannol, avenanthramide-2c, avenanthramide-2f, dihydromyricetin, hypericin, mangiferin, and verbascoside (Supplementary Fig. 16a). The detailed inhibitory potency of all compounds was shown in Supplementary Fig. 16B. As TrxR has attracted increasing interests as a promising anticancer drug target[10,58], the easy synthesis of Fast-TRFS and ready availability of crude tissue extracts make this method particularly useful in discovering TrxR inhibitors. With an automatic operating system, this method is readily extended to a middle- or high-throughput screening assay.

## Discussion

A well-established but the structurally undefined strategy for controlled release of target molecules is the CDR process[22–32]. Here we disclosed the structural factors determining the rate of such process by rational design and evaluation of a series of TrxR probes (TRFS1-8). The leaving group (X in Fig. 1b), the ring size of the resulting cyclization product (n in Fig. 1b) and the nucleophilic group (Z in Fig. 1b) were systematically examined. Clarification of the structural determinants of the CDR process will have a broad interest, and further shed light on the construction of different stimuli-responsive molecules. After we examined the properties of TRFS1-8, TRFS5 was identified to have a fast response rate towards TrxR. However, the following experiments demonstrated that the selectivity of TRFS5 was compromised as it was also activated by GSH (Table 2 and Supplementary Fig. 2A). Thus we next turned to optimize the recognition part of the probe. After studying the interaction of cyclic disulfides/diselenides with TrxR, we concluded here that 5-membered cyclic disulfides were selectively reduced by TrxR, which laid a foundation for developing chemical tools specifically targeting TrxR.

Another interesting and unexpected discovery is that we observed a fluorescence off-on signal by a direct cleavage of disulfide/diselenide bonds (TRFS3, TRFS6, TRFS8, Fast-TRFS, TRFS10, and TRFS12), which has not been reported before.

Theoretical calculations indicated that the non-emissive property of Fast-TRFS is due to the occurrence of a photo-induced electron transfer (PET) process within the excited molecules ($S^2 \rightarrow S^1$ transition), while there is no such PET process in the excited R-Fast-TRFS. The cleavage of the disulfide bond without a slow CDR process contributed to the fast response and improved selectivity of the probe to TrxR. It has long been known that reduction of the disulfide bond in oxidized Trx proteins may restore the emission of the tryptophan residue adjacent to the disulfide bond, and this observation was explained by the conformational difference between the reduced proteins (no disulfide bond) and the oxidized proteins (with a disulfide bond close to the tryptophan residue)[59]. Alternatively, we may readily account for the observed gain of fluorescence is due to the direct cleavage of the disulfide bond in the oxidized Trx proteins.

The elevated TrxR activity in tumor tissue is critical to maintain the phenotypes of cancer cells, such as fast DNA synthesis, unlimited proliferation and resistance to apoptosis, and thus inhibition of TrxR has been considered as a therapeutic approach for cancer treatment[10,60]. Mammalian TrxR enzymes are selenoproteins that contain a Sec residue in their C-terminal active sites. The Sec is incorporated cotranslationally at a UGA codon, normally read as a stop codon, for protein synthesis in ribosomes. It remains a challenge to prepare recombinant selenoproteins from bacterial expression systems as the translation usually stops at the site of Sec incorporation[61,62], which limits the availability of selenoproteins. The favorable properties of Fast-TRFS prompted us to develop a convenient and economic screen assay to discover TrxR inhibitors using crude tissue extracts as a source of TrxR, bypassing the requirement of the purified selenoenzyme. This plate reader-based method is easily extended to a high-throughput screen assay, thus greatly facilitating the large-scale discovery of TrxR inhibitors.

In summary, Fast-TRFS, a superfast and specific fluorescent probe of TrxR with a different sensing mechanism, has been discovered. The response rate was improved by dissecting the structural determinants of the CDR process, and the high specificity was achieved by exploring the selectivity of cyclic disulfides/diselenides reduction by TrxR. Disclosing the SAR of the CDR process has a general interest and would advance the development of different probes, prodrugs, and theranostic agents. In addition, clarifying the selective reduction of the 1, 2-dithiolanes by TrxR defines a general scaffold for constructing chemical tools specifically targeting TrxR. The sensing mechanism of Fast-TRFS, i.e., switching on the fluorescence by a direct cleavage of the disulfide bond, also suggests that disulfide/diselenide bonds may quench the fluorescence signal of certain fluorophores, and thus a cleavage of disulfide/diselenide bonds could serve a trigger in fluorescent probe design. Finally, the multiple favorable properties of Fast-TRFS enabled to develop a convenient and economic method to screen TrxR inhibitors using crude tissue extracts as a source of TrxR, and dozens of natural inhibitors of TrxR have been identified.

## Methods

**Materials and instruments**. The recombinant rat TrxR1, with a specific activity of 50% of the wild TrxR1 with the DTNB assay, was a gift from Prof. Arne Holmgren at Karolinska Institute, Sweden. The recombinant U498C TrxR1 mutant (Sec → Cys) was produced as described[63]. Dulbecco's modified Eagle's medium (DMEM), glutathione (GSH), dimethyl sulfoxide (DMSO), yeast glutathione reductase (GR) and auranofin (AF) were obtained from Sigma-Aldrich (St. Louis, USA). NADPH was obtained from Roche (Mannheim, Germany). Fetal bovine serum (FBS) was obtained from Sijiqing (Hangzhou, China). Aniti-TrxR1 antibody (sc-28321, 1:2000 dilution) and anti-mouse IgG-HRP (sc-2031, 1:4000 dilution) were purchased from Santa Cruz Biotechnology (Santa Cruz, USA). All organic solvents and starting materials for organic synthesis are of analytical grade and were purchased from commercial supplies. The absolute quantum yields (ϕ) of Fast-TRFS with and without TCEP were determined on FLS920 spectrometer (Edinburgh Instruments,

U.K.). [1]H and [13]C NMR spectra were recorded on Bruker Advance 400 or Varian 400, and tetramethylsilane (TMS) was used as a reference. MS spectra were recorded on Trace DSQ GC-MS spectrometer or Bruker Daltonics esquire 6000 mass spectrometer or Shimadzu LCMS-2020. HRMS was obtained on Orbitrap Elite (Thermo Scientific). HPLC analysis were performed on Shimadzu LCMS-2020 system with a Wondasil C18 Superb reversed-phase column (5 μm, 4.6 × 150 mm). The column was eluted with methanol and water. The flow rate was set at 0.6 mL min$^{-1}$. A PDA detector or a mass detector was used to monitor the products. The compounds in the library used for screen were purchased or synthesized. Celastrol, guggulsterone, bufalin, embelin, tanshinone IIA, eupalinolide A, gossypol, decursin, morroniside, isoliquiritigenin, cardamonin, huperzine A, morin, magnolol, honokiol, diosmetin, limonin, rhein, crocin were purchased from Chengdu Biopurify Phytochemicals Ltd., Chengdu, China; http://www.biopurify.cn. Cynaropicrin, gambogic acid, cynarin, harpagoside, fisetin, mangiferin were purchased from Baoji Chenguang Biotechnology Co., Ltd., Baoji, China; http://www.herbest. cn. 7-Ketocholesterol, cortisone, prednisone, dexamethasone, curcumin, piperine, disulfiram, ibrutinib, exemestane, caffeic acid, ferulic acid, resveratrol, piceatannol were purchased from J&K China Chemical Ltd., China; http://www.jkchemical. com. Thymoquinone, sophocarpine, santonin, dehydroleucodine, eupalinilide B, piperlongumine, caffeic acid phenethyl ester, myricetin, cichoric acid, lipoic acid, lipoamide, hydroxytyrosol, 6-gingerol, chlorogenic acid, dihydromyricetin, obacunone, silibinin were purchased from Shanghai Yuanye Biotechnology Co., Ltd., Shanghai, China; http://www.shyuanye.com. Securinine, shikonin, alantolactone, baicalein, artemisinin, dihydroartemisinin, hypericin, verbascoside were purchased from Pufei De Biotechnology Co., Ltd., Chengdu, China; http://www.sc-victory. com. Plumbagin, parthenolide, and D3T were purchased from Santa Cruz Biotechnology (Santa Cruz, CA). Xanthatin was purchased from ChemFaces, Wuhan, China; http://www.chemfaces.cn. Asparagusic acid was synthesized according to the literature[64]. Xanthohumol was synthesized according to our previous publication[65]. Avenanthramide-2c, avenanthramide-2f, avenanthramide-2p were synthesized according to the literature[66]. The detailed synthesis and characterization of different cyclic diselenides/disulfides and TRFS series probes were presented in the Supplementary Information.

**Absorbance and fluorescence spectroscopy**. UV−vis spectra were acquired from an UV−vis spectrometer (Evolution 200, Thermo Scientific). Fluorescence spectroscopic studies were performed on a Cary Eclipse Fluorescence Spectrophotometer (Agilent Technologies). The slit width was 5 nm for both excitation and emission. For spectra measurements, TRFS series probes were dissolved in DMSO to obtain a stock solution, which were diluted with Tris (50 mM)/EDTA (1 mM) buffer (TE buffer, pH 7.4) to the desired concentrations.

**Cell lines and culture conditions**. HeLa cells were obtained from the Shanghai Institute of Biochemistry and Cell Biology, Chinese Academy of Sciences, and were authenticated by Shanghai Biowing Biotechnology Co. LTD (Shanghai, China). The cells were kept in DMEM with 10% FBS, 2 mM glutamine and 100 units mL$^{-1}$ penicillin/streptomycin, and maintained in a humidified atmosphere of 5% CO$_2$ at 37 °C. HeLa-sh*NT* and HeLa-sh*TrxR1* cells were generated in our lab and were cultured under the same conditions as those of HeLa cells with additional supplement of puromycin (1 μg mL$^{-1}$) in DMEM[16,18].

**Live cell imaging**. HeLa cells, HeLa-sh*NT* cells, and HeLa-sh*TrxR1* cells were grown as described. When the cells reached ~70% confluence, they were treated as described in the corresponding figure legends. The images were acquired on FLoid Cell Imaging Station (Thermo Fisher) under the blue channel.

**[1]H NMR spectra of Fast-TRFS and R-Fast-TRFS**. Fast-TRFS (3.8 mg, 0.01 mmol) was dissolved in 500 μL of DMSO-d$_6$. TCEP (29 mg, 0.1 mmol) was dissolved in 500 μL of TE buffer. The two solutions were mixed to give a 1:1 solution. Then the mixed solution was incubated in the dark at 37 °C. After 2 h, the solution was diluted with 20 mL of distilled dichloromethane. The organic phase was washed with water (3×10 mL), dried over Na$_2$SO$_4$ and evaporated to dryness in vacuo. The residue was redissolved in 500 μL of DMSO-d$_6$ and analyzed by nuclear magnetic resonance (NMR) spectroscopy.

**Theoretical calculations**. All calculations were performed using Gaussian 16 software package. Geometry optimizations of ground state molecules were carried out at ωB97X-D/Def2-SVP/IEFPCM(water) level of theory[67–69], the *i*th singlet excited state (*i* = 1 or 2) structures were optimized at TD-ωB97X-D/Def2-SVP/IEFPCM(water) (nstates = *i* + 3, root = *i*) level of theory, and the UV-Vis absorption/emission spectra were carried out at TD-ωB97X-D/Def2-SVP/IEFPCM (water) (nstates = 20, root = *i*) level of theory. Vibrational frequencies were also calculated to verify the optimized structures are energy minima, and no imaginary frequency was found. The optimized structures are shown using CYLview[70], and the calculated absorption/emission spectra were shown using Multiwfn software[71].

**Reporting summary**. Further information on research design is available in the Nature Research Reporting Summary linked to this article.

## Data availability

The source data underlying Figs. 2, 4, 5, 6 and 7, Table 3 and Supplementary Figures 1, 2, 8, 9, 10, 11, 12, 13, 14 and 16 were provided as a Source Data file. Other data are available from the corresponding authors upon reasonable request.

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

## Acknowledgements

The financial supports from the National Natural Science Foundation of China (21572093 & 21778028), the Natural Science Foundation of Gansu Province (18JR4RA003) and the 111 project were greatly acknowledged. We appreciated Prof. Jincai Wu (Lanzhou University) for his valuable discussion on fluorescence sensing mechanisms of Fast-TRFS.

## Author contributions

X.L. and J.F. designed and initiated the study and J.F. directed the project. X.L., B.Z., J.L., S.W., X.W. and X.J. performed the organic synthesis, spectroscopic studies, LC-MS studies, cell experiments and screening experiments. C.Y. and P.Z. performed the theoretical calculations. X.L. C.Y., and J.F. prepared the manuscript.

## Additional information

**Competing interests:** The authors declare no competing interests.

