## [Peer Review File · Nature Communications]

Reviewers' comments:

Reviewer #1 (Remarks to the Author):

The manuscript by Fang and coworkers, "Dissecting structural determinants leading to discovery of a superfast and specific fluorescent probe of thioredoxin reductase with unprecedented sensing mechanism" concerns the development of a new continuous assay for TrxR based on previous work by the same authors.

I think the work presented in the manuscript is quite useful to the field as the authors present the development of a new probe that allows for a fluorescent signal to be generated once a disulfide substrate is cleaved. The innovation of this work is that the new probe, Fast-TRFS does not undergo cyclization driven release, unlike previous probes. Apparently, the disulfide moiety of the probe quenches the fluorophore. Once the disulfide is reduced to the dithiol, this quenching is relieved and a signal is produced. This overcomes the previous problem where CDR was slow in the other probes this group developed.

I am generally supportive of the work here, but the major problem is that this manuscript is MUCH, MUCH too long. Readers are not going to be interested in all of the structure-function details that led them to develop this new probe. The authors need to figure out a new way to say the same thing in about half of the space. It was a laborious read and will be uninteresting to many readers because of all of the detail in this manuscript.

A second problem is that the authors presented some basic facts incorrectly and omitted citation of key papers. On page 12 of the manuscript the authors state that: "However, studies on the interaction of cyclic disulfides with TrxR are limited."

They cite a few papers but miss one very important paper by Lothrop and coworkers (Lothrop AP, Ruggles EL, Hondal RJ. (2009) No selenium required: reactions catalyzed by mammalian thioredoxin reductase that are independent of a selenocysteine residue. *Biochemistry*. 48(26):6213-23. doi: 10.1021/bi802146w.)

The paper by Lothrop shows that TrxR reduces the disulfide of lipoic acid (5-membered ring), but not 1,2 dithiane rings (like the oxidized form of DTT - 6-membered rings). They conclude, like Singh and Whitesides (a paper the authors DID cite) that ring strain of the 5-membered disulfide contributed greatly to it being able to be reduced by TrxR. Strained disulfides are electrophilic and thus can be reduced by TrxR as the study by Lothrop shows. Unstrained disulfides are unreactive because they are not electrophilic.

This brings me to another point that the authors got wrong. On page 13 of the manuscript, the authors state: "Second, the interactions of GSH/TrxR with disulfides/diselenides involve a general thiol/selenol and disulfides/diselenides exchange reaction, which is ubiquitous in biological systems and forms a fundamental of the biological redox regulation. There are two major factors, i.e. the nucleophilicity of the attacking group (thiolate or selenolate) and the stability of the resulting leaving group (thiolate or selenolate), to determine such exchange reactions."

The authors cite a very important paper (#52) by Koppenol and coworkers. The important thing that Koppenol showed was that the two most important factors in thiol/disulfide exchange or selenol/disulfide exchange are nucleophilicity of the attacking nucleophile and the ELECTROPHILICITY of the atom accepting the electrons from the nucleophile. The leaving group ability of a selenolate and a thiolate were comparable. In fact, Koppenol showed that electrophilicity of the selenium atom in a selenosulfide contributed 100-fold more to rate acceleration compared to the nucleophilicity of the attacking selenolate. This fact was recently reinforced by O'Keefe and coworkers (O'Keefe JP, Dustin CM, Barber D, Snider GW, Hondal RJ. (2018) A "Seleno Effect" Differentiates the Roles of Redox Active Cysteine Residues in Plasmodium falciparum Thioredoxin Reductase. *Biochemistry*. 57(11):1767-1778.

Once the authors greatly shorten the manuscript and revise the facts above, I will be happy to review a revised manuscript.

Reviewer #2 (Remarks to the Author):

In this manuscript authors developed fast-TRFS as a specific and superfast fluorogenic probe with a stable -NH-C(O)-NH- linker for mammalian thioredoxin reductase (TrxR). Authors constructed many probes by changing the different linker units, sulfide/selenide atoms in the ring and fluorophores and studied their sensing properties. Authors provided the comparative response rate data of TRFS-green and TRFS-red with Fast-TRFS. The specificity of Fast-TRFS to TrxR and live cell imaging in HeLa shNT cells and HeLa-shTrxR1 cells were studied.

Although they developed many probes with stable -NH-C(O)-NH- linker units and tested their sensing property without CDR process, authors failed to provide and explain with clear evidence for why probes are getting fluoresced without fluorophore release in its original form. Moreover, the novelty of the work and studies did not meet the "Nature communications" standards. Hence, I recommend that this work may be published in some other journal not in the "Nature communications".

Reviewer #3 (Remarks to the Author):

Fang's group developed a fluorescent probe that detects thioredoxin reductase activity with fast response and high selectivity. They explored the optimal structure of disulfide substrates, fluorophores and linker units for the reductase probe and found that a five-membered disulfide, a coumarin fluorophore and a urea linker are ideal structure of the selective and fast detection of the enzymatic activity. The probe named FAST-TRFS enhanced fluorescence immediately after mixing with the enzyme and showed the selectivity. No significant response to other intracellular thiols such as glutathione was observed. These properties are attractive for sensing the reductase activity in living cells. Furthermore, the probe seems to function with a new sensing mechanism. This is somewhat surprising, because the coumarin is ICT-type fluorophore that needs a donor functional group for fluorescence emission, but it still keeps a urea structure at the 7th position, which is electron-withdrawing. Although the sensing mechanism is chemically interesting, I have a strong concern in terms of practical use of the probe. The excitation wavelength of the probe is 345 nm in the UV range, which is very toxic to cells. Autofluorescence is also a problem even in cell-lysate-based experiments as well as live-cell imaging. In experiments for inhibitor screening, there will be a high possibility of excitation of inhibitors, resulting in undesirable fluorescence emission. While the study is thoroughly conducted, this may not be so appealing to the readership of nature communications. However, as I mentioned, the sensing principle is new and interesting. Thus, this paper will be more suitable for journals specialized in the field of chemistry. To strengthen the chemical value, it will be important to make the mechanism clearer. Some comments below will be helpful for enhancing the chemical quality of the paper.

1. Is the absorption spectrum changed after reduction of the disulfide bond? How is the NMR spectrum changed? I am wondering if the electron drawing urea structure should be somehow changed for emission. Is there a possibility of intramolecular nucleophilic attack of the reduced thiol to the carbonyl group of the urea to form an hemithioacetal structure, which is not electron withdrawing and triggers fluorescence emission.

2. For selectivity experiments, glutathione is used in 1 mM. But the maximum level of intracellular glutathione is reported to be 10 mM.

Point-by-point Response to Reviewers**Reviewer #1 (Remarks to the Author):**

The manuscript by Fang and coworkers, "Dissecting structural determinants leading to discovery of a superfast and specific fluorescent probe of thioredoxin reductase with unprecedented sensing mechanism" concerns the development of a new continuous assay for TrxR based on previous work by the same authors.

I think the work presented in the manuscript is quite useful to the field as the authors present the development of a new probe that allows for a fluorescent signal to be generated once a disulfide substrate is cleaved. The innovation of this work is that the new probe, Fast-TRFS does not undergo cyclization driven release, unlike previous probes. Apparently, the disulfide moiety of the probe quenches the fluorophore. Once the disulfide is reduced to the dithiol, this quenching is relieved and a signal is produced. This overcomes the previous problem where CDR was slow in the other probes this group developed.

I am generally supportive of the work here, but the major problem is that this manuscript is MUCH, MUCH too long. Readers are not going to be interested in all of the structure-function details that led them to develop this new probe. The authors need to figure out a new way to say the same thing in about half of the space. It was a laborious read and will be uninteresting to many readers because of all of the detail in this manuscript.

A: According to the reviewer's suggestion, we have shortened the structure-function details of the probes. We briefly kept the conclusions in the manuscript, and all the detailed description and interpretation of the results were moved to the Supporting Information.

A second problem is that the authors presented some basic facts incorrectly and omitted citation of key papers. On page 12 of the manuscript the authors state that: "However, studies on the interaction of cyclic disulfides with TrxR are limited."

They cite a few papers but miss one very important paper by Lothrop and coworkers (Lothrop AP,

Ruggles EL, Hondal RJ.(2009) No selenium required: reactions catalyzed by mammalian thioredoxin reductase that are independent of a selenocysteine residue. *Biochemistry*. 48(26):6213-23. doi: 10.1021/bi802146w.) The paper by Lothrop shows that TrxR reduces the disulfide of lipoic acid (5-membered ring), but not 1,2 dithiane rings (like the oxidized form of DTT - 6-membered rings). They conclude, like Singh and Whitesides (a paper the authors DID cite) that ring strain of the 5-membered disulfide contributed greatly to it be able to be reduced by TrxR. Strained disulfides are electrophilic and thus can be reduced by TrxR as the study by Lothrop shows. Unstrained disulfides are unreactive because they are not electrophilic.

This brings me to another point that the authors got wrong. On page 13 of the manuscript, the authors state: "Second, the interactions of GSH/TrxR with disulfides/diselenides involve a general thiol/selenol and disulfides/diselenides exchange reaction, which is ubiquitous in biological systems and forms a fundamental of the biological redox regulation. There are two major factors, i.e. the nucleophilicity of the attacking group (thiolate or selenolate) and the stability of the resulting leaving group (thiolate or selenolate), to determine such exchange reactions." The authors cite a very important paper (#52) by Koppenol and coworkers. The important thing that Koppenol showed was that the two most important factors in thiol/disulfide exchange or selenol/disulfide exchange are nucleophilicity of the attacking nucleophile and the ELECTROPHILICITY of the atom accepting the electrons from the nucleophile. The leaving group ability of a selenolate and a thiolate were comparable. In fact, Koppenol showed that electrophilicity of the selenium atom in a selenosulfide contributed 100-fold more to rate acceleration compared to the nucleophilicity of the attacking selenolate. This fact was recently reinforced by O'Keefe and coworkers (O'Keefe JP, Dustin CM, Barber D, Snider GW, Hondal RJ. (2018) A "Seleno Effect" Differentiates the Roles of Redox Active Cysteine Residues in Plasmodium falciparum Thioredoxin Reductase. *Biochemistry*. 57(11):1767-1778.

A: We appreciate the reviewer's comments, and are sorry for the incorrect interpretation and missing an important reference. Accordingly, we have corrected the expression in the revised manuscript (Page 10, text with a yellow background), and the relevant references by O'Keefe et al. and Lothrop et al. were included in the revised manuscript (Refs 44 & 47).

Once the authors greatly shorten the manuscript and revise the facts above, I will be happy to review a revised manuscript.

A: We appreciate the reviewer's helpful comments, and have shortened the manuscript and addressed the concerns raised by the reviewer.

Reviewer #2 (Remarks to the Author):

In this manuscript authors developed fast-TRFS as a specific and superfast fluorogenic probe with a stable -NH-C(O)-NH- linker for mammalian thioredoxin reductase (TrxR). Authors constructed many probes by changing the different linker units, sulfide/selenide atoms in the ring and fluorophores and studied their sensing properties. Authors provided the comparative response rate data of TRFS-green and TRFS-red with Fast-TRFS. The specificity of Fast-TRFS to TrxR and and live cell imaging in HeLa shNT cells and HeLa-shTrxR1 cells were studied.

Although they developed many probes with stable -NH-C(O)-NH- linker units and tested their sensing property without CDR process, authors failed to provide and explain with clear evidence for why probes are getting fluoresced without fluorophore release in its original form. Moreover, the novelty of the work and studies did not meet the "Nature communications" standards. Hence, I recommend that this work may be published in some other journal not in the "Nature communications".

A: We have performed theoretical calculations, and provided evidence to clarify the off-on sensing mechanism of the probe upon reduction (page 14, text with a green background). For the probe itself, the energy difference between two excited states (S^1 and S^2) is low (0.22 eV), which enables electrons in the first excited state (S^2) to be transferred to another excited state (S^1) via intersystem crossing (ISC). Thus, the probe displays extremely low fluorescence. Upon reduction, there is no ISC process in the excited states of the reduced probe. When the electrons return from the excited state of the reduced probe (S^1) to the ground state, light is emitted and is seen as fluorescence. The detailed results of calculation and interpretation of the results were given in the revised text (page 14, text with a green background) and Supporting Information.

Regarding the novelty, we think that there are at least 4 novel discoveries in this work:

First, the cyclization-driven release (CDR) is widely applied in designing controlled release molecules, such as probes, prodrugs and theranostic agents. We disclosed the general structural

determinants of the CDR process, which would advance the construction of novel controlled release systems. Second, discovery of small molecule ligands of a protein of interest is critical for chemical manipulation of the protein, but it remains a high challenge. We disclosed 1, 2-dithiolans as highly selective ligands of mammalian thioredoxin reductase (TrxR), defining a general scaffold for constructing novel chemical tools specifically targeting TrxR. Third, a drastic increase of fluorescence of the probe upon cleavage of the disulfide bond disclosed an unprecedented sensing mechanism, which not only accounts for some experimental observations (e.g., higher fluorescence intensity for reduced thioredoxin proteins than oxidized thioredoxin proteins) but also advances the design of novel probes. Forth, we discovered a superfast and highly specific fluorescent probe of TrxR, which would facilitate the exploration of TrxR functions under physiological and pathological conditions. Thus, we believe the novelty of this work meets the standard of Nature communications.

Reviewer #3 (Remarks to the Author):

Fang's group developed a fluorescent probe that detects thioredoxin reductase activity with fast response and high selectivity. They explored the optimal structure of disulfide substrates, fluorophores and linker units for the reductase probe and found that a five-membered disulfide, a coumarin fluorophore and a urea linker are ideal structure of the selective and fast detection of the enzymatic activity. The probe named FAST-TRFS enhanced fluorescence immediately after mixing with the enzyme and showed the selectivity. No significant response to other intracellular thiols such as glutathione was observed. These properties are attractive for sensing the reductase activity in living cells. Furthermore, the probe seems to function with a new sensing mechanism. This is somewhat surprising, because the coumarin is ICT-type fluorophore that needs a donor functional group for fluorescence emission, but it still keeps a urea structure at the 7th position, which is electron-withdrawing.

Although the sensing mechanism is chemically interesting, I have a strong concern in terms of practical use of the probe. The excitation wavelength of the probe is 345 nm in the UV range, which is very toxic to cells. Autofluorescence is also a problem even in cell-lysate-based experiments as well as live-cell imaging. In experiments for inhibitor screening, there will be a high possibility of excitation of inhibitors, resulting in undesirable fluorescence emission. While the study is thoroughly

conducted, this may not be so appealing to the readership of nature communications. However, as I mentioned, the sensing principle is new and interesting. Thus, this paper will be more suitable for journals specialized in the field of chemistry. To strengthen the chemical value, it will be important to make the mechanism clearer. Some comments below will be helpful for enhancing the chemical quality of the paper.

A: We understand the concern of the reviewer regarding the off-on sensing mechanism of the probe as the coumarin scaffold is a typical ICT-type fluorophore. We presented solid evidence in the original manuscript to identify the product after the probe is reduced. To further understand the off-on sensing mechanism of the probe, we performed additional theoretical calculations of the probe (Fast-TRFS) and its reduced product (R-Fast-TRFS). For the probe itself, the energy difference between two excited states (S^1 and S^2) is low (0.22 eV), which enables electrons in the first excited state (S^2) to be transferred to another excited state (S^1) via intersystem crossing (ISC). Thus, the probe displays extremely low fluorescence. Upon reduction, there is no ISC process in the excited states of R-Fast-TRFS. When the electrons return from the excited state of R-Fast-TRFS (S^1) to the ground state, light is emitted and is seen as fluorescence. The detailed results of calculation and interpretation of the results were given in the revised manuscript (page 14, text with a green background) and Supporting Information. Another concern from the reviewer is the practical use of the probe. Although the probe has a blue emission, there is no problem for the probe to be applied for live cell imaging as the probe has a large F/FO signal, which was demonstrated in the manuscript. In the cell lysate-based screen of TrxR inhibitors, we monitored the time-course change (e.g., kinetics) of the fluorescence signal, which may bypass the possible interference from the background fluorescence signals of samples. As we disclosed a novel sensing mechanism in this work, we are continuing this project, and hope to report probes with longer emission wavelength in due time.

This work deals with several fundamental questions in chemistry, biochemistry and medicinal chemistry, such as dissecting structural determinants of the cyclization-driven release (CDR) process, finding specific ligands of a target protein, creating a specific fluorescent probe and developing a convenient high-throughput screening assay for identifying TrxR inhibitors. Thus, we think this work has a broad readership, and is suitable for Nature communications.

1. Is the absorption spectrum changed after reduction of the disulfide bond? How is the NMR spectrum changed? I am wondering if the electron drawing urea structure should be somehow changed for emission. Is there a possibility of intramolecular nucleophilic attack of the reduced thiol to the carbonyl group of the urea to form an hemithioacetal structure, which is not electron withdrawing and triggers fluorescence emission.

A: There is no apparent change of the absorption spectrum after reduction of the disulfide bond (Page 12, text with a yellow background & Fig. S12). There are remarkable changes of protons' signal from the dithiolane ring after reduction of the disulfide bond (Fig. S13, SI). Accordingly, these results were included in the revised text (Page 14, text with a yellow background). We appreciate the reviewer's suggestion on the off-on sensing mechanism of the probe. However, based on our results (Fig. 5A), there is only one quantitative product (e.g., R-Fast-TRFS) after the probe is reduced, and no hemithioacetal structure product is observed.

2. For selectivity experiments, glutathione is used in 1 mM. But the maximum level of intracellular glutathione is reported to be 10 mM.

A: According to the reviewer's suggestion, the response of the probe with 10 mM of GSH was determined. As shown in (Page 15, text with a yellow background & Fig. S14), little fluorescence signal was observed. Taken together with the results from the TrxR knockdown experiments (Fig. 5F, Fig. 6C & Fig. 6E), we may conclude that Fast-TRFS is a highly selective probe of TrxR.

REVIEWERS' COMMENTS:

Reviewer #1 (Remarks to the Author):

My critiques have been adequately addressed.

Reviewer #2 (Remarks to the Author):

In the revised manuscript Prof Fang and colleagues have been explained R-Fast-TRFS getting fluoresced without fluorophore release in its original form by Frontier molecular orbital theory. And other comments, they have been addressed satisfactorily. Hence, I recommend that this manuscript will accept for publication.

1. In Fig. 4, should be modify to "Response of TRFS9, TRFS10 & TRFS 12 to TCEP" instead of "TRFS9-13 to TCEP".

Reviewer #3 (Remarks to the Author):

The authors revised the paper and responded to my comments. The chemical mechanism of the fluorescence modulation was described using the results of quantum chemical calculation. The other questions regarding spectral analyses and glutathione effects were also addressed. Although the excitation wavelength, rather than the emission wavelength, is short for live-cell imaging, practical use of the probe was answered. As I mentioned the novelty of chemical mechanism of the coumarin emission, I can recommend the acceptance of the publication of the paper in nature communications.

Point-by-point Response to Reviewers

Manuscript #: NCOMMS-19-01786A

REVIEWERS' COMMENTS:

Reviewer #1 (Remarks to the Author):

My critiques have been adequately addressed.

Response: We appreciate the reviewer's comments, and are happy that the reviewer is satisfied with our revision.

Reviewer #2 (Remarks to the Author):

In the revised manuscript Prof Fang and colleagues have been explained R-Fast-TRFS getting fluoresced without fluorophore release in its original form by Frontier molecular orbital theory. And other comments, they have been addressed satisfactorily. Hence, I recommend that this manuscript will accept for publication.

1. In Fig. 4, should be modify to "Response of TRFS9, TRFS10 & TRFS 12 to TCEP" instead of "TRFS9-13 to TCEP".

Response: We appreciate the reviewer's comments, and are happy that the reviewer is satisfied with our revision. The legend of Figure 4 has been revised to correct the inappropriate expression.

Reviewer #3 (Remarks to the Author):

The authors revised the paper and responded to my comments. The chemical mechanism of the fluorescence modulation was described using the results of quantum chemical calculation. The other questions regarding spectral analyses and glutathione effects were also addressed. Although the excitation wavelength, rather than the emission wavelength, is short for live-cell imaging, practical use of the probe was answered. As I mentioned the novelty of chemical mechanism of the coumarin emission, I can recommend the acceptance of the publication of the paper in nature communications.

Response: We appreciate the reviewer's comments, and are happy that the reviewer is satisfied with our revision.